# The Brain at High Altitude: From Molecular Signaling to Cognitive Performance

**DOI:** 10.3390/ijms241210179

**Published:** 2023-06-15

**Authors:** Mostafa A. Aboouf, Markus Thiersch, Jorge Soliz, Max Gassmann, Edith M. Schneider Gasser

**Affiliations:** 1Institute of Veterinary Physiology, Vetsuisse Faculty, University of Zürich, 8057 Zurich, Switzerland; 2Department of Biochemistry, Faculty of Pharmacy, Ain Shams University, Cairo 11566, Egypt; 3Zurich Center for Integrative Human Physiology (ZIHP), University of Zurich, 8057 Zurich, Switzerland; 4Institute Universitaire de Cardiologie et de Pneumologie de Québec (IUCPQ), Faculty of Medicine, Université Laval, Québec, QC G1V 4G5, Canada; 5Neuroscience Center Zurich, University of Zurich and ETH Zurich, 8057 Zurich, Switzerland

**Keywords:** hypoxia, HIF, EPO, neurogenesis, synaptogenesis, carotid body, acclimatization, cognition

## Abstract

The brain requires over one-fifth of the total body oxygen demand for normal functioning. At high altitude (HA), the lower atmospheric oxygen pressure inevitably challenges the brain, affecting voluntary spatial attention, cognitive processing, and attention speed after short-term, long-term, or lifespan exposure. Molecular responses to HA are controlled mainly by hypoxia-inducible factors. This review aims to summarize the cellular, metabolic, and functional alterations in the brain at HA with a focus on the role of hypoxia-inducible factors in controlling the hypoxic ventilatory response, neuronal survival, metabolism, neurogenesis, synaptogenesis, and plasticity.

## 1. High Altitude and Cognition

High altitude (HA) (defined as an altitude above 2500 m above sea level) is characterized by multiple harsh environmental conditions. Most physiological adaptations occur in response to the reduced atmospheric pressure, resulting in reduced oxygen partial pressure and causing reduced blood oxygenation saturation (SpO_2_), hypoxemia. The brain is susceptible to alterations in oxygen supply. Thus, HA exposure causes adverse changes in mood states, such as depression [1] and anxiety [2], and neurocognitive alterations, such as memory impairment [3] and attention disorders after both short- and long-term HA exposure [4,5]. Although numerous reports concern the physiological and neurological alteration that occurs after ascent to HA, less has been investigated on the cognitive and brain alterations in long-term and permanent inhabitants of HA.

Brain functions are affected by hypoxia not only after ascent to HA [6] but also after long-term exposure at a HA [7] and in native highlanders [8]. In unacclimatized individuals exposed to HA, sleep patterns can already be affected at elevations above 1600 m, changes in mood states like euphoria or depression are observed in some individuals from 2500 m on, and above 3000 m subjects can experience headache, dizziness, and confusion. Mood state alterations, including euphoria, quarrelsome, irritability, and apathy, occur temporarily after fast acute exposure to HA and return to baseline states after 48 to 52 h [9,10,11]. In contrast, short- and long-term exposure to HA causes biological, inflammatory, and structural brain changes that increase the risk of experiencing anxiety and depression symptoms [12] and neurocognitive dysfunctions such as slower reaction times, reduced attention (>3500 m), impaired learning, spatial and working memory (>4000 m), and impaired retrieval (>5500 m) (Figure 1) [7,8,13,14].

Mood alterations after HA exposure are attributed to changes in brain levels of dopamine and serotonin. Euphoria is attributed to increased brain dopamine levels [15], while lower serotonin levels are attributed to sadness, grief, worry, anxiety, and depression [16]. Studies in rodents and piglets have shown increased extracellular dopamine levels in the striatum when the oxygen pressure in the cerebral cortex decreases [17,18]. Moreover, reduced serotonin (5-hydroxytryptamine, 5-HT) availability is observed at HA in people with anxiety and depression, particularly in women [19]. Long-term exposure to HA induces a hypoxic stress response in which the noradrenergic brain regions, particularly the locus coeruleus and hypothalamic-pituitary-adrenal (HPA) axis, are involved [20]. Similarly, HA leads to alterations in brain bioenergetics, which could also contribute to depressive symptoms. Indeed, multiple magnetic resonance spectroscopy studies suggest that persons at HA with depression exhibit alteration in adenosine triphosphate (ATP) expression [21]. Similarly, an imbalance in mitochondrial dynamics has been demonstrated in the rat brain hippocampus after hypobaric hypoxia exposure [22]. Moreover, HA residents (1400 m, Salt Lake City in Utah, USA) exhibit significant differences in brain pH and inorganic phosphate levels compared with sea-level residents (13 m, Belmont City in Massachusetts, USA) [23].

Acute, subacute, and repeated HA exposure causes a reduction in neurocognitive processing speed (reaction time) and attention (switching, visual processing). A study in healthy subjects showed that acclimatization improves reaction time and attention due to increased SpO_2_. Thus, processing speed and attention are associated with SpO_2_ acclimatization [13].

Despite about 2.2% of the world’s population (more than 140 million people) living permanently at a HA [24], the impact of reduced SpO_2_ on cognition remains poorly explored. SpO_2_ values decrease with increasing altitude to a median of 96% (95–97) at 2500 m, 92% (90–93) at 3600 m, 87% (85–89) at 4100, and 81% (78–84) at 5100 m, with increasing variability at higher altitudes [25]. The reduced SpO_2_ at HAs induces a rapid increase in hypoxic ventilatory response (HVR) [26], but also breathing instability (periods of deep and rapid breathing) and central apnea during sleep [27]. Although the rate of sleep apnea is lower at HA than at sea level [28], the gap in SpO_2_ between wakefulness and sleep states is greater than at sea level [29,30]. The statistical distribution of SpO_2_ in children at HA shows that one out of four children saturates significantly less than the others [31]. Moreover, in one out of four children at HA, the oxygen desaturation index during sleep (which reflects SpO_2_ variability) is lower [32,33]. Changes in sleep patterns can modulate mood and cause deficits in working memory, attention, or executive functions [34]. Yet, whether low SpO_2_ during sleep correlates with impaired cognition at HA remains unknown.

Acute exposure to HA causes impairment in learning and memory encoding and retention [35]. Those alterations are also observed in lowlanders living at HA [36]. With the development of cognitive neuroscience research techniques such as functional magnetic resonance imaging (fMRI) and event-related potentials (ERP), it is possible now to explore the mechanisms underlying the effects of long-term HA exposure on cognition. After natural selection, highlanders have genetic and physiological adaptations to HA. Thus, brain alterations in HA-exposed lowlanders may not reflect the brain of adapted highlanders. Exposure to HA affects mainly visual spatial attention in lowlanders [37]. After the ascent to very HA, brain edema and cortical atrophy were reported in climbers [38], but the reduction in brain volume was only reported after more than three weeks of very HA exposure (>5500 m) [39]. For HA immigrants, an increase in gray matter is observed, probably associated with increased neurogenesis and vasculature. Alterations in white matter volumes after acute HA exposure were rarely observed, but diffuse imaging suggested injury [40]. Highlander descendants of Han immigrants showed increased interhemispheric fibers [41]. The changes in cerebrospinal fluid were very variable across studies [42].

fMRI is used to measure changes in blood dynamics caused by neuronal activity. Blood oxygenation level-dependent (BOLD) images and ERP studies showed a reduction in oxygenation and activity in the occipital visual cortex, insular cortex, cingulate cortex, precentral gyrus, hippocampus, and cerebellum in individuals staying at HA or HA natives [42].

Electroencephalogram (EEG) is used to evaluate the electrophysiological processing of cognitive activity, mostly after ascent to HA and in HA immigrants, but studies in HA natives are rare. In general, EEG shows that prolonged exposure to HA mainly impairs attention. In native adolescents in Bolivia, reductions in delta and beta frequency amplitude were recorded [43]. In Tibetan natives, a higher executive ability and impaired orienting ability were observed only at HA above 4000 m, suggesting a threshold for the influence of HA in brain function [7].

In highlanders living above 4000 m, lower levels of performance in executive functions are observed [8]. We have also measured a reduction in spatial, abstract, and verbal abilities in adolescents living in El Alto, Bolivia (4150 m). HA significantly increases the risk for neurodevelopmental deficits, with larger effects on females [44]. On the contrary, attention deficit hyperactive disorder (ADHD) is less prevalent with increasing altitude, proposing a HA protective effect [45]. Thus, controversy is related to the possible neurocognitive function impairment at HA. However, the corresponding studies conducted so far have the drawback that they do not control biases associated with the socioeconomic level and family, school, and social environments.

In summary, highlanders have adapted to hypoxia through different physiological mechanisms (reviewed in [46]), such as increased lung capacity, increased ventilation, improved oxygen diffusion, increased total vessel density [47], and higher hemoglobin concentration. However, at very HA, impaired cerebral dynamic in blood flow [48], reduced cerebral glucose metabolism [49], reduced cerebral vascular reactivity [50,51], altered gray and white matter [42], and altered neuronal activity are some of the alterations that may affect neurocognitive functioning [52]. Understanding the molecular mechanisms underlying the brain adaptation to HA provides insights into developing interventions to mitigate the adverse effects of HA exposure, such as acute mountain sickness (AMS) and HA cerebral edema (HACE). This review article provides an overview of the molecular changes in the brain in response to HA-induced hypoxia, focusing on neurotransmitter systems, signaling pathways, and neuronal metabolism. We will also discuss the implications of these findings for HA-related diseases and potential avenues for therapeutic intervention.

## 2. Hypoxia-Inducible Factors (HIFs) in the Brain Adaptation to HA

At HA, low oxygen supply leads to changes in gene expression mediated by the hypoxia-inducible factors (HIFs) pathway, which play a key role in the cellular response to hypoxia. In 2019, the Nobel Prize in Physiology was awarded to William Kaelin, Peter Ratcliffe, and Gregg Semenza for the discovery of HIFs as master regulators of oxygen homeostasis [53]. HIFs are heterodimers composed of an oxygen-sensitive alpha-subunit (HIF-α) and a constitutive beta-subunit (HIF-1β, also known as ARNT, aryl hydrocarbon receptor nuclear translocator). The two major HIF-α isoforms are HIF-1α and HIF-2α. Although the kinetics of stabilization and transactivation of both isoforms are similar [54], recent genome-wide studies have identified that the *EPAS1* gene that encodes for HIF-2α is the major isoform involved in HA adaptation [55]. *EPAS1* distribution is tissue- and cell-specific, mainly expressed in organs such as the kidney [56,57] and brain [58].

Hypoxic HIF-2α stabilization at HA leads to the formation of the HIF-2 complex that activates the expression of erythropoietin (EPO). EPO is a glycoprotein hormone that, when released from the kidney, enhances the formation of red blood cells (erythropoiesis) in the bone marrow to increase the blood’s oxygen-carrying capacity [59]. In the brain, EPO regulates the neural respiratory zones (central and peripheral), leading to higher HVR, thereby increasing tissue oxygenation [26]. EPO also increases brain angiogenesis (formation of new vessels) [3,60], neurogenesis (formation of new neurons) [3], synaptogenesis (new synapses) [61], and brain oxidative metabolism and cognition [62,63] (Figure 2).

On the other hand, the HIF-1 complex controls the expression of vascular endothelial growth factor (VEGF), an important signaling protein involved in vasculogenesis, angiogenesis (reviewed in [64]), and neurogenesis [65]. After 11 days of ascent to a HA, we observed that inhibition of VEGF signaling in rats interferes with vasculogenesis, angiogenesis, and neurogenesis. In contrast, EPO mainly involves HA-mediated angiogenesis in non-neurogenic brain areas [3] (Figure 2). In Sherpa highlanders, where tolerance to hypoxia is partly attributed to an increase in microcirculatory blood flow and capillary density, the VEGFA plasma levels did not correspond to HA and remained equivalent to the level in non-Sherpa low landers. This finding was speculated to be associated with distinctive genetic variations in the promoter region of *VEGFA* [66]. Conversely, an increase in VEGFA and EPO in the Andean population is related to chronic mountain sickness (CMS) [67].

HIF-1 also controls the expression of brain-derived neurotrophic factor (BDNF) and insulin-like growth factor-I (IGF-I), both key growth factors that regulate neurogenesis and synaptogenesis from embryonic to adult stages (reviewed in [68,69]). BDNF also interacts with astrocytes and neurons to control respiration [70,71,72]. Animal and human research indicate that physical activity can enhance BDNF blood levels and gene expression [73]. In contrast, long-term exposure to hypoxia lowers BDNF serum levels [74], yet the impact of HA on BDNF expression remains unexplored. No changes in IGF-I expression were observed after acute or chronic exposure to HA [75].

HIF-1 and -2 are crucial for proper brain development and continuously expressed in adult brain neurogenic zones and neural stem/progenitor cells from the embryonic and postnatal mouse brain [76]. HIF-1α deficient mice exhibit hydrocephalus accompanied by reduced neuronal cells and spatial memory impairment [77]. Additionally, HIF-2 has been shown to protect neural progenitor cells and neural differentiation processes by upregulating the survival orthologues Birc5a and Birg5b during embryogenesis [78]. Moreover, recent studies showed that loss of HIF-2α can affect cognitive performance in mice and causes a loss of pyramidal neurons in the retrosplenial cortex, a brain area responsible for spatial navigation [58]. This specific role of HIF-2 in neural development and synaptic plasticity suggests that it is crucial for cognitive function. Thus, targeting HIF-mediated modifications may hold therapeutic potential for HA-related illnesses such as AMS, HACE, and CMS neurologic deficits.

## 3. Signaling Pathways Interacting with HIFs

Neuronal viability is maintained through a complex interacting network of signaling pathways that can be disturbed in response to many cellular stresses. An imbalance in these signaling pathways after stress or in response to pathology can have drastic consequences for the function or fate of neurons. The mechanisms underlying the HIF-mediated changes in the brain involve a complex interplay between various signaling pathways, including the cyclic adenosine monophosphate (cAMP) pathway, phosphoinositide 3-kinase (PI3K) pathway, and nitric oxide (NO) pathway.

**The cAMP pathway** is an important second messenger system that plays a critical role in regulating numerous neural processes in the brain, from development, cellular excitability, synaptic plasticity, learning, and memory (reviewed in [79,80,81]). HIF-1 activates the cAMP pathway in response to hypoxia in cancer cells [82,83]. In the brain, cAMP leads to increased expression of the N-methyl-D-aspartate (NMDA) receptor subunit GluN1 in the hippocampus, an area involved in learning and memory [84]. cAMP can also regulate the activity of various downstream effectors involved in synaptic plasticity and memory formation, such as the protein kinase A (PKA) [85] and the cAMP response element-binding protein (CREB) [86]. Moreover, the cAMP signaling pathway can modulate other molecular targets in the brain, such as mitochondrial function and oxidative stress, by activating CREB and the peroxisome proliferator-activated receptor gamma coactivator-1 alpha (PGC-1α) [87]. A longitudinal cohort study of proteomic and clinical biomarkers of AMS symptom phenotypes in 53 individuals after ascent to HA showed that the cAMP pathway interferes with the symptoms of AMS [88]. Therefore, this pathway could be a potential therapeutic strategy for AMS and understanding acclimatization to HA.

**The Wnt/β-catenin pathway** is directly regulated by HIF and is involved in developing brain structures such as the cerebral cortex and hippocampus (reviewed in [89]). The key mediator of Wnt signaling, the armadillo protein-β-catenin, participates in transcriptional regulation and chromatin interactions and is regulated by hypoxia [89]. The Wnt-mediated activation of HIF-1 promotes the maintenance of embryonic and neural stem cell activity and is significantly decreased in differentiated cells [90]. Furthermore, chronic exposure to hypoxia in vivo induces activation of the Wnt/β-catenin signaling cascade in the hippocampus, suggesting that mild hypoxia may have therapeutic value in neurodegenerative disorders [91].

**The PI3K/Akt pathway** is another important signaling pathway critical in regulating cell growth and survival and has been widely reported in brain development, aging, neurodegenerative diseases, and psychotic disorders (reviewed in [92]). HIF-1 and PI3K/Akt may interact on functional and regulatory levels. PI3K/Akt is required for heat shock proteins to protect HIF-1α from Von-Hippel-Lindau (pVHL)-independent degradation [93]. However, another earlier study has refuted this dependent interaction [94]. In a rat model of Alzheimer’s disease, it has been shown that gamma-aminobutyric acid (GABA) type B receptor-mediated PI3K/Akt activation alleviates oxidative stress and neuronal cell injury [95]. Moreover, in cerebral ischemic injury, the co-activation of GABA_A_ and GABA_B_ receptors exerted a neuroprotective effect via the PI-3K/Akt pathway [96]. Also, activation of the PI3K/Akt pathway has been shown to have neuroprotective effects and improve cerebral blood flow in animal models of cerebral ischemia [97]. Therefore, targeting this pathway could be a potential therapeutic strategy to treat or prevent HACE. One study demonstrated that treating hypoxic mouse brain microvascular endothelial cells with a PI3K activator, 3-methyladenine, suppressed hypoxia-induced endothelial permeabilization [98] and might effectively tackle HACE. Increased Akt activation (and overexpression of Parkin—a molecule that plays a critical role in ubiquitination) has been shown to reduce hypoxia-induced death of induced pluripotent stem cells-derived neurons from CMS. It is proposed that increased Akt activation protects against hypoxia-induced cell death. Therefore, it is suggested that impaired adaptive mechanisms, including lack of Akt activation and increased Parkin expression, render neurons from chronic mountain sickness subjects more susceptible to hypoxia-induced cell death [99].

**The NO signaling pathway** plays a significant role in HA-related illnesses, particularly in regulating vascular function and blood flow (reviewed in [100]). Hypoxia-induced HIF activation leads to increased expression of endothelial NO synthase (eNOS) [101]. NO acts as a vasodilator, causing the relaxation of vascular smooth muscle cells, increasing blood flow to oxygen-deprived tissues, and maintaining endothelial homeostasis [102]. NO also plays a pivotal role in regulating neurotransmitter release, such as acetylcholine, catecholamines, and neuroactive amino acids [103], as well as synaptic plasticity in the cerebral cortex [104]. Alterations in NO signaling have been implicated in several HA-related illnesses, including pulmonary edema and HACE [100]. In HA pulmonary edema, impaired NO production and increased pulmonary vasoconstriction can lead to increased fluid accumulation in the lungs. Tibetans have NO levels in the lung, plasma, and red blood cells that are at least double and, in some cases, orders of magnitude greater than in other populations, regardless of altitude [100]. Therapeutic interventions targeting the NO signaling pathway, e.g., administering NO donors or phosphodiesterase inhibitors, have shown promise in preventing and treating HA-related illnesses [105]. HIF-2-regulated EPO induced NO release in endothelial cells [106], cardiomyocytes [107], and lung cancer cells [108]. NO, in turn, co-regulated mitochondria and energy metabolism in combination with Akt. A similar mechanism was witnessed in the murine postnatal hippocampus, corresponding to enhanced cognition in early adulthood [63]. Moreover, NO (and derived S-nitrosothiols and glutathionylation) is required for the physiological response to hypoxia. NO directly interacts with brainstem respiratory centers and modulates the ventilatory response to hypoxia [109]. Furthermore, the current findings suggest that NO’s activity mediates the excitatory and inhibitory components of the hypoxic ventilatory response, and that NO may play a role in modulating the prominent second phase of the biphasic response to hypoxia [110].

**Glutathionylation** is crucial in redox signaling and cellular adaptation to oxidative stress. It is an important mechanism in regulating the activity of various proteins, including ion transporters and hypoxia-inducible factor 1 (HIF-1) pathway proteins. Sodium/Potassium ATPase (Na+/K+-ATPase) is an essential membrane protein responsible for maintaining the electrochemical gradient across the plasma membrane of cells. Hypoxia-induces glutathionylation of specific cysteine residues in the α-subunit of Na+/K+-ATPase. This modification can alter the activity of the pump, affecting ion balance and cellular function [111,112]. Also, glutathionylation of sarcoplasmic/endoplasmic reticulum (ER) calcium ATPase (SERCA) occurs under hypoxia, leading to altered calcium handling and ER stress [113]. Glutathionylation can also affect the DNA-binding activity of HIF-1α. It has been suggested that glutathionylation of specific cysteine residues in HIF-1α may enhance its binding to the hypoxia response elements (HREs) present in the promoter regions of target genes, thereby influencing their transcriptional activation [114]. Overall, glutathionylation of proteins, including ion transporters and HIF-1, is a crucial mechanism in the cellular response to hypoxia. It allows cells to modulate the activity of proteins involved in ion balance, calcium signaling, and gene expression, enabling adaptation and survival under conditions of oxygen limitation. Further research to fully understand the specific cysteine residues targeted for glutathionylation and the functional consequences of this modification on protein activity in the context of hypoxia is needed.

## 4. Central Respiratory Acclimatization to HA

The most important feature of acclimatization to HA is the increase in depth and rate of breathing. Ventilatory acclimatization to hypoxia occurs days to weeks after hypoxia exposure and is characterized by progressive augmentation of ventilation until reaching a plateau. Ventilatory acclimatization is completed for a given hypoxic stimulus in humans and rodents after 7–21 days [115,116]. The ventilatory acclimatization to hypoxia depends on the peripheral chemoreceptors: the carotid bodies [117,118]. Carotid bodies contain the glomus cells type I, in which HIF-2α is expressed [119]. Mice with a partial knockout of HIF-1α and HIF-2α exhibit an impaired ventilatory response [115,120].

EPO is a key molecule of the HVR, acting as a respiratory stimulant on both the central nervous system (CNS) (brainstem) and carotid bodies [26]. EPO attenuates hypoxia-induced respiratory depression (a decline in the respiratory bursting activity), activating the extracellular-signal-related kinase (ERK) and protein kinase B (Akt) pathways [121]. The impact of EPO on the neural control of ventilation is sex-specific at both the brainstem and carotid body levels [122]. Under hypoxia, EPO enhances respiratory frequencies in males but higher tidal volumes in females. A similar sex-specific effect on the impact of EPO in ventilation has been observed in adult men and women receiving intravenous injections of EPO [122].

Carotid body glomus cells contain O_2_- and CO_2_-sensitive K+ channels, which are inhibited by hypoxia or hypercapnia [123]. The inhibition of K+ channels leads to cell depolarization, Ca2+ entry, and the release of neurotransmitters and ATP [124,125]. Such increased neurotransmitter release stimulates the carotid sinus nerve activity, leading to increased ventilation in response to hypoxia and hypercapnia [126]. The glomus cells also synthesize NO, which is not stored in vesicles but functions as a chemical messenger that inhibits the excitation induced by hypoxia and, consequently, the ventilatory response [125]. By performing ex vivo recordings of the carotid sinus nerve, we showed a dual effect of EPO in response to hypoxia and hypercapnia. EPO concentrations within a specific concentration range (0.1–0.5 IU/mL) stimulate sensory activity in response to hypoxia but not hypercapnic stimuli. Conversely, EPO concentration exceeding 1 IU/mL decreased the sensory responses to hypoxia and hypercapnia, apparently due to an exacerbated production of NO [127].

## 5. The Neuronal Response to HA

Information on the (human) neuronal response to HA exposure is rare. Most information has been obtained from animal models, i.e., predominantly hypoxia-intolerant species such as rats. These models were exposed to hypoxia in chambers mimicking HA conditions. However, most studies investigated neuronal responses after acute, intermittent, or prolonged normobaric hypoxia, not in chronic hypobaric hypoxia at HA. In the following paragraphs, we reviewed and discussed the impact of HA and hypoxia on (i) neurogenesis, (ii) neuronal survival, (iii) neuronal metabolism, (iv) neurotransmitter release, and (v) neuronal membrane potential.

### 5.1. Neurogenesis

Neurogenesis in the adult mammalian brain occurs primarily in two regions, the dentate gyrus (DG) of the hippocampus and the subventricular zone (SVZ) of the lateral ventricles. Both SVZ and DG stem cell niches are slightly hypoxic (reviewed in [128]), which supports the survival and maturation of stem or progenitor cells [129,130]. Mechanistically, both HIF-1 and HIF-2 have been reported to regulate neurogenesis. Loss of HIF-1α ultimately impairs adult neurogenesis [131], indicating that HIF-1 promotes neurogenesis [132] and contributes to maintaining the neural stem cell pool [133]. Likewise, the loss of HIF-2α function in neurospheres reduced neuronal differentiation [134]. Moreover, EPO upregulation also stimulates neurogenesis [135], further suggesting a role for HIF-2 in neurogenesis [136]. In addition to endogenous or functional hypoxia, prolonged (few hours to days) exposure to hypoxia and HA increases neurogenesis in rats [3,137] and mice [91] also via VEGF [3]. Thus, it may be possible that HIF-1 and HIF-2 act synergistically in neurogenesis. Neural stem cells in the SVZ of rats were reported to resist prolonged severe hypoxia at 8–10% ambient oxygen levels (corresponding to altitudes of 6000 m and higher) [138]. However, such extreme conditions resulted in structural disorganization of the germinal center and apoptosis of neurons and oligodendrocytes [138]. In contrast, rats exposed to hypobaric hypoxia showed decreased neurogenesis with increasing duration of hypoxia exposure [139], suggesting that prolonged exposure to HA may rather suppress neurogenesis. Thus, the neurogenesis may be activated or suppressed depending on oxygen partial pressure and the duration of exposure.

### 5.2. Neuronal Survival

Impaired cognitive functions at HA indicate that neuronal function and survival are reduced during prolonged exposure to HA. Rodents chronically exposed (8 weeks at 4300 m [140]; 21 days at 6100 m [141]) to hypobaric hypoxia show cognitive deficits associated with increased neuronal apoptosis in the cortex, striatum, and hippocampus. Reduced blood perfusion, increased demyelination, and microglia activation correlate with neuronal deficits during chronic adaptation [142]. With time, however, rodents (rats) recover some cognitive functions and show reduced neuronal cell death in at least some areas, such as the cortex or the cornu ammonis (CA1) [141]. Thus, the neuronal network seems to stabilize after acclimatization, and acute rather than chronic hypoxia exposure causes neuronal degeneration. The mechanism of apoptosis-stimulating effects after acute brain hypoxia is best studied in hypoxia/ischemia (H/I) models, which stimulates neuronal apoptosis better than hypoxia exposure alone [143]. H/I induces neuronal apoptosis in neonatal [144] and adult rodents [145] by activating a complex signaling network. Neuronal apoptosis is executed by activation of Bcl-2 family members, such as BAX and BAK [146,147,148], and caspase 3 [144,149], 8 [150], and 9 [151]. Activation of neuronal apoptosis after H/I is regulated by several pathways, including signaling through NFkB [152,153,154,155,156,157], GSK-3b [121], PI3K-AKT [158,159,160,161], RAS/MEK/ERK [162], JNK [163,164], or AMPK/Foxo3a [165,166,167,168]. The role of HIFs in neural apoptosis and survival at HA has yet to be fully analyzed. Upregulation of HIF-1α expression precedes activation of caspases during hypoxia [143] and H/I [143,169]. Increased HIF-1α levels reduce neuronal death (by increasing VEGF [170] or EPO expression [171], for instance) when activated during the early phase after H/I. Increased neuronal apoptosis after ascent to HA in rats can also be prevented with an enriched environment-mediated increase of VEGF and EPO [3]. While HIF-1 protected neurons in vitro against hypoxia, HIF-1α upregulation in astrocytes promoted neuronal apoptosis [172], suggesting cell-type specific functions of HIF-1 during H/I. We showed that HIF-1 in photoreceptors was not required for neuronal protection after hypoxic preconditioning [173], although the simultaneous activation of HIF-1 and HIF-2 in normoxic VHL-deficient mice protected against photoreceptor apoptosis [174]. However, other groups reported that the downregulation of HIF-1α in the neurons of the brain is associated with decreased neuronal apoptosis [175,176], suggesting that HIF-1α may even promote neuronal apoptosis [177,178]. Indeed, the loss of HIF-1α in neurons protected against H/I damage [179]. A recent study suggested that HIF-1 and HIF-2 contribute to neuronal apoptosis and that the loss of either HIF-α subunit can be compensated by the other α subunit [180]. Zhang et al. [92] argued that the duration and severity of hypoxia determine the pro- or anti-apoptotic functions of HIF-1 and possibly of HIF-2 in neurons. During short (30 min) hypoxia, HIF-1 contributes to neuronal damage [179]. In contrast, after longer (75 min) exposure, HIF-1 protected neurons from apoptosis and decreased tissue damage [181]. The role of HIFs in neuronal protection against hypoxia at HA has not yet been investigated. HIF-1 (and possibly also HIF-2) may protect neurons during prolonged hypoxia [92], suggesting that HIFs may contribute to the protection of neurons during prolonged exposure (days to weeks) to hypobaric hypoxia. We recently analyzed the retinal response to hypobaric hypoxia at 3500 m (corresponding to 13–14% oxygen). We observed no neurodegeneration and apoptosis, although the outer photoreceptor segments were shortened in HA-exposed mice [182]. It may be possible that HIFs in the retina protect against hypoxia at HA [183,184]. Furthermore, we showed that FVB line mice exposed to 10% hypoxia for 6 h had higher HIF-1α expression in the brainstem than Prague Dawley rats exposed to the same conditions [185]. This suggests that mice and rats respond differently to hypoxia, supporting the hypothesis that mice have innate traits that favor adaptation to HA. Studies testing the role of HIFs during acute exposure, after acclimatizing, and in adapted organisms are needed to understand their role in neuronal damage in hypobaric hypoxia at HA.

### 5.3. Neuronal Metabolism

Neurons have a high energy demand to maintain neural network activities, ionic gradients, and membrane excitability. Thus, neurons need a high oxygen supply to generate sufficient levels of ATP. Under normoxic conditions, neurons produce energy through aerobic and anaerobic metabolism [186,187]. During hypoxia, oxidative phosphorylation ceases, and ATP levels drop rapidly [188,189]. The resulting energy deficit can be temporarily compensated by activating glycolysis for the anaerobic production of ATP [189,190,191]. The glycolytic capacity may determine neuronal survival during prolonged hypoxia [192]. Indeed, increased glucose levels protect neurons from damage during hypoxia in hippocampal slices [193,194]. However, glycogen storage in the brain is limited and prolonged hypoxia rapidly drains brain glucose [195,196]. Additionally, the accumulating end products of glycolysis, such as lactate, and the acidification of the tissue microenvironment by protons (H+) may harm the neurons and limit the reliance on glucose metabolism [195]. After hypoxia, brain glycogen is partially regenerated [197], and lactate is removed via the bloodstream [195,198] or catabolized into pyruvate to re-enter the tricarboxylic acid cycle [195,197].

In Quechuas, Andean highlanders living between 3500 and 4900 m, glucose metabolism is reduced in several brain regions, especially regions with higher cortical functions [199]. The hypometabolism in these adapted highlanders may be a mechanism to protect against chronic hypoxia [199], preventing decreases in tissue pH, which can impede synaptic functions [200,201]. In contrast to Quechuas, the glucose metabolism in the brain of Tibetan Sherpas from the Himalayas did not differ from that of lowlanders [202]. The unchanged glucose metabolism in Tibetans suggests they are better adapted to HAs and may need less protection against chronic hypoxia. Of note, the metabolic adaptation of neurons seems to differ between highlanders from South America, Tibet, and potentially Ethiopia. For example, Ethiopian Amhara highlanders are protected against hypoxia, probably due to increased cerebral blood flow and increased oxygen delivery to the brain [203]. Caucasian lowlanders exposed to HA for a prolonged time showed an increased glucose metabolism in the cerebellum but a decreased glucose metabolism in the frontal cortex, left occipital cortex, and the thalamus [204], which may be a hypoxia defense mechanism with similarities to the mechanism in Quechuas. These similarities suggest that the brain metabolism in Quechuas reflects an earlier stage of adapting the brain metabolism to HA. Most probably, neurons in Quechua, Tibetan, and Aymara people developed independent strategies to adapt metabolically to HA.

Many HA- or anoxia-adapted species suppress neuronal function by the neurotransmitter GABA [205] to limit energy consumption [206]. Naked mole rats (*Spalax*) live and survive in nearly anoxic conditions. Yet, their brain tolerates exposure to severe hypoxia [207,208] or in vitro ischemia [209,210], probably due to the sustained expression of the NMDA receptor subunit GluN2D [211]. Decreasing ATP levels and the resulting energy deficit during hypoxia are often associated with an excessive release of glutamate from neurons [212,213,214]. The neurotransmitter homeostasis of glutamate, aspartate, and GABA is coupled to the mitochondrial metabolism because intermediates of the tricarboxylic acid cycle serve as precursors for neurotransmitter synthesis [196]. Astrocytes feed glutamine to neurons [215] (which convert glutamine into glutamate or GABA) and clear glutamate from synapses after neuronal release [196]. Glutamate helps to maintain the energy balance, but increased glutamate levels drive excitotoxicity, neuronal cell death caused by excitatory amino acids, in the brain, especially during HI [140,196]. Non-adapted, hypoxia-susceptible species lower the brain levels of glutamine but not glutamate during hypoxia. In contrast, naked mole rats show reduced glutamate and glutamine levels during hypoxia, which may protect the brain better against excitotoxicity [216]. During hypoxia, naked mole rats also rewire the glycolysis to drive the pentose phosphate pathway (PPP) [216,217]. Glucose metabolism via PPP is 17% less efficient than glycolysis [196], but the PPP is essential for nucleotide and lipid synthesis and nicotinamide-adenin-dinucleotid-phosphat (NADPH) regeneration [196]. NADPH is required to regenerate glutathione to counteract reactive oxygen species, which rise especially after hypoxic episodes [46]. In vitro experiments suggest a tight regulation between PPP and glycolysis in neurons [218] and that the PPP contributes to neuronal resistance to severe hypoxia [208]. HIF-1α and HIF-2α are constitutively expressed at high levels in various tissues of hypoxia-tolerant species like *spalax,* even under normoxic conditions. This elevated baseline expression is believed to preserve redox balance and energy metabolism even in low-oxygen environments [219].

Species not adapted to HA can sustain severe hypoxia only briefly because they cannot suppress their metabolic rate and, thus, their ATP demand. A reduced metabolism can only be achieved by lowering the body temperature [196]. However, repetitive exposure to hypoxia allows mice to enter a hypometabolic state with decreased body temperature and brain oxygen consumption. Additionally, the expression of two key glycolytic enzymes, phosphofructokinase and pyruvate kinase, and oxidative phosphorylation are suppressed in brains repetitively exposed to hypoxia [220]. Thus, it may even be possible that hypoxia-intolerant species can activate mechanisms to survive prolonged episodes of hypoxia. Indeed, hypoxia-acclimatized rodents showed increased cerebral blood flow [221], glucose uptake [222], and metabolism [223] but decreased cytochrome c [224]. This suggests that the mitochondrial metabolism is suppressed during hypoxia. The suppression of mitochondrial metabolism can persist for several hours after exposure to hypoxia [225]. During severe hypoxia, mitochondrial function can be irreversibly impaired in species not adapted to hypoxia [226]. Mitochondria depolarize and start leaking ions and apoptosis-inducing factors. In the secondary injury after ischemia, mitochondria produce reactive oxygen species (ROS) and are targeted by ROS during reoxygenation after episodes of severe hypoxia [227], especially the accumulation of succinate controls ROS production and reperfusion injury [228]. During reoxygenation, succinate is oxidized to fumarate by the mitochondrial complex II (succinate dehydrogenase) of the electron transport system, driving a massive generation of ROS [206]. Anoxia-tolerant species manage succinate levels by preventing succinate production during hypoxic episodes [229] or redirecting the accumulated succinate [230]. The mitochondria of hypoxia-resistant naked mole rats show high elasticity. Naked mole rats can reversibly reduce their metabolic rate [216], brain oxygen consumption by 85%, and mitochondrial function by 90% in acute hypoxia [231]. If exposure to HA results in succinate accumulation in humans and how HA-adapted highlanders or other species manage succinate are, to our knowledge, not known.

Likewise, the exact mechanisms by which hypoxia affects ROS generation in neurons at HA are not yet fully understood and may vary depending on the duration and severity of the hypoxic exposure. Research on animals and humans exposed to HA has demonstrated changes in antioxidant enzyme activities and antioxidant molecule levels in the brain [232]. These adaptations are believed to be an essential part of the cellular response to counteract the increased ROS production and maintain neuronal function. Neurons are particularly vulnerable to oxidative stress due to their high metabolic activity, high content of oxidizable fatty acids, and relatively low levels of antioxidant enzymes compared with other cell types [233]. However, neurons have developed an antioxidant system to counterbalance ROS and maintain cellular redox balance. Some of the key components of the neuronal antioxidant system include several antioxidant enzymes, such as superoxide dismutase (SOD), catalase, and glutathione peroxidase; non-enzymatic antioxidants to scavenge ROS, including molecules such as vitamin C, vitamin E, glutathione, and uric acid; and various transcription factors, including nuclear factor erythroid 2-related factor 2 (Nrf2) that controls the expression of multiple antioxidant genes and helps maintain redox homeostasis [234].

### 5.4. Neurotransmitter Release

In the brain, HIF regulates the expression of neurotransmitter systems, which are important for maintaining proper brain function. Here, we will discuss the role of HIF in regulating neurotransmitter systems and the molecular mechanisms underlying these effects.

**Glutamate** is the most abundant excitatory neurotransmitter in the brain [235]. Hypoxia can alter glutamate signaling by affecting the expression of glutamate receptors and transporters in HIF-dependent [236] and HIF-independent ways [237]. In HIF-1α conditional knockout mice, H/I induces elevation of extracellular glutamate and NMDAR activation [238]. On the other hand, hypoxia is reported to suppress glutamate uptake into astrocytes via activation of NF-κB but not HIF. Suppression of glutamate uptake may be an important factor in HI-triggered glutamate excitotoxicity [239].

**GABA**, the primary inhibitory neurotransmitter in the brain, is synthesized by glutamic acid decarboxylase (GAD) enzymes in the inhibitory neuron and, thus, the deregulation of GAD enzymes and subsequent change in GABAergic activity is involved in various neurological and neuropsychiatric diseases [240]. Hypoxia can alter GABA signaling by affecting the expression of GABA receptors and transporters. GABA type A (GABA_A_) receptors are ligand-gated chloride channels that mediate fast inhibitory transmission, whereas GABA type B receptors (GABA_B_) are metabotropic receptors that produce prolonged inhibitory signals via G proteins and second messengers [241]. One study evaluated in vivo GABA_B_ receptor alterations and gene expression changes in glutamate decarboxylase and HIF-1α in the cerebral cortex of hypoxic neonatal rats and glucose, oxygen, and epinephrine in the resuscitation groups. Under hypoxic stress, a significant decrease in total GABA and GABA_B_ receptors, GABA_B_, and GAD expression is observed in the cerebral cortex, which accounts for respiratory inhibition. At the same time, HIF-1α was upregulated under hypoxia to maintain body homeostasis [242]. In addition, hypoxia can lead to decreased expression of the GABA transporter GAT-1 in the cortex, leading to altered GABA levels. Another study examined the expression of HIF-1α in the different neuronal phenotypes under in vitro and in vivo ischemia and showed that the HIF-1α expressing GAD65/67-positive interneurons also possessed high levels of glutathione, which might explain their longer survival under cerebral ischemia [243,244]. Neuronal HIF-1α was reported to be a necessary signal for ventilatory acclimatization to hypoxia and the plasticity in glutamatergic neurotransmission in the nucleus tractus solitarius in the CNS with chronic hypoxia [245].

**Dopamine** is a neurotransmitter involved in regulating motivation, reward, and movement [246]. Hypoxia can alter dopamine signaling by affecting the expression of dopamine receptors and transporters. HIF regulates the expression of the dopamine transporter DAT, which is responsible for clearing dopamine from the synapse. Under hypoxic conditions, HIF decreases the expression of DAT, leading to increased dopamine levels in the synapse and may contribute to HA-related illnesses such as AMS. HIF prolyl hydroxylase inhibition augments dopamine release in the rat brain in vivo [247] and in vitro [248]. Chemical-induced up-regulation of HIF-1α was shown to prevent dopaminergic neuronal death via the activation of MAPK family proteins in chemically induced neurodegenerative mice [249] and enhanced the dopaminergic phenotype and neurite outgrowth via upregulating tyrosine hydroxylase and dopamine transporter by the nuclear receptor estrogen-related receptor γ [250]. On the other hand, the profound difference between the induction patterns of type D3 and D4 postsynaptic dopamine receptors DRD3 and DRD4 and the direct HIF-1α target gene *VEGFA* implies that the DRD3 and DRD4 promoters might not be activated directly by HIF-1α. Still, other slow-reacting hypoxia-sensitive transcription factors might be involved in their transcriptional regulation [251].

**Serotonin** and its receptors are important in regulating virtually all brain functions, and dysregulation of the serotonergic system has been implicated in the pathogenesis of many psychiatric and neurological disorders [252]. Hypoxia can alter serotonin signaling by affecting the expression of serotonin receptors and transporters [253,254]. HIF regulates the expression of the serotonin transporter SERT, which is responsible for clearing serotonin from the synapse. Under hypoxic conditions, HIF decreases the expression of SERT, which leads to increased serotonin levels in the synapse and may contribute to HA-related illnesses such as HACE. The HIF-serotonin signaling axis promotes axon regeneration in *C. elegans* through the serotonin receptor SER-7 that activates the cAMP signaling pathway, ultimately promoting the neuron regenerative response. This serotonin synthesis activation is mediated by HIF-1α [255].

**Acetylcholine (ACh)** functions as an important neurotransmitter in the autonomic nervous system, CNS, and at the neuromuscular junction. Chronic hypoxia (10% oxygen), as short as 15 min, decreases ACh synthesis in adult and developing rat brains [256]. In the carotid body, hypoxia has a biphasic effect on ACh release, i.e., initial facilitation followed by sustained inhibition [257]. Another study showed that the synthesis of Ach in rats, when measured via glucose labeling, decreased by 35 and 54% with 15% oxygen and 10% oxygen, respectively. Still, when measured via choline labeling, it was reduced by 50 and 68% with 15% oxygen and 10% oxygen, respectively [258]. A recent study reported that in-utero hypoxia attenuated ACh-mediated vasodilatation in the ovine middle cerebral artery [259]. HIF-2 can regulate ACh signaling in the brain by modulating the expression of choline acetyltransferase, which is the enzyme that synthesizes ACh. HIF-2 activation has also been shown to increase the expression of nicotinic ACh receptors, which play a role in cognitive function and memory.

In summary, HIF plays a critical role in regulating neurotransmitter systems in the brain in response to hypoxia. By controlling the expression of neurotransmitter transporters, HIF helps to maintain proper neurotransmitter levels in the synapse and protects against excitotoxicity or over-inhibition. However, dysregulation of HIF-mediated neurotransmitter signaling can contribute to HA-related illnesses and affect brain function.

### 5.5. Neuronal Membrane Potential and Ion Exchange

Neurons under acute and chronic hypoxia alter their membrane resting potential and conductance. O_2_-sensitive potassium (K^+^) channels (i.e., Ca^2+^-activated (KCa), two pore domain (TWIK)-related acid-sensitive (TASK), inward-rectifying (Kir), and ATP-sensitive (KATP)) play a critical role in regulating membrane conductance [260]. Under hypoxia, the membrane from excitatory neurons hyperpolarizes, associated with an outward leak current and increased resting conductance at the holding potential [261]. These effects are not influenced by inward flow of chloride ions. Thus, hypoxia increases the resting conductance in excitatroy neurons. Hypoxia also activates KATP in inspiratory neurons in the brainstem, controlling their membrane resting potential [262]; moreover, hypoxic activation of Kir7.1 in oligodendrocytes maintains cell integrity and stimulates the process of myelination [263].

Neurons from hypoxia-tolerant animals, such as the anoxia-tolerant turtle (*Chrysemys picta*), survive without oxygen for hours. A hypothesis for survival is metabolic depression by “channel arrest”. This consists of a reduction of leakage throughout the membrane during anoxic episodes. The decrease in leaks results in conserved energy by reducing the demand for ion pumps [264]. There are no significant changes in membrane resistance in cortical neurons from the anoxia-tolerant turtle during short-term anoxia, indicating that the channel arrest defense mechanism is not utilized for energy conservation. Instead, an anoxia-mediated decrease in whole cell NMDA receptors and α-amino-3-hydroxy-5-methyl-4-isoxazolepropionic acid receptor (AMPAR) currents are an important part of the turtle’s natural defense to reduce glutamatergic excitability. Mitochondrial ATP-sensitive K(+) (mK(ATP)) channels participate in the reduction of AMPA currents during anoxia, being a common mechanism in the protection of excitotoxicity [265].

## 6. HIF-2 and Synaptic Plasticity

HIF-2 plays a pivotal role in neural plasticity, including changes in synapse formation, dendritic morphology, and long-term potentiation (LTP). Neural plasticity refers to the brain’s ability to modify its structure and function in response to experiences and environmental changes. One important aspect of neural plasticity is the formation and modification of synapses, the communication sites between neurons. LTP is a form of synaptic plasticity that underlies learning and memory and involves strengthening synaptic connections between neurons [266]. HIF-2 has been shown to regulate the expression of genes involved in synapse formation and function, such as BDNF and postsynaptic density protein 95 (PSD-95) via EPO [267,268]. Furthermore, HIF stabilization by dimethyloxallyl glycine-mediated inhibition of prolyl hydroxylase domain enzymes (PHDs) has been implicated in LTP in rat hippocampus [269]. Studies have also demonstrated the involvement of HIF-2 in dendritic morphology, which refers to dendrite structure, the branched extensions of neurons that receive incoming signals from other neurons. HIF-2 has been shown to promote dendritic growth and branching in response to hypoxia [270]. HIF-2 can also regulate post-translational modifications of proteins in neurons, including phosphorylation, acetylation, and ubiquitination. For example, HIF-2 can induce the phosphorylation of the CREB transcription factor [271] involved in learning and memory under hypoxia and even under atmospheric oxygen conditions. Stabilized HIF-2α protein was found in the brains of adult mice. Neuro-specific HIF-2α-knockout mice showed a reduction of pyramidal neurons in the retrosplenial cortex, a brain region responsible for a range of cognitive functions, including memory and navigation. Accordingly, behavioral studies showed disturbed cognitive abilities in these mice. This specific cellular loss was attributed to deficits in migration in neural stem cells from HIF-2α knockout mice due to altered expression patterns of genes highly associated with neuronal migration and positioning [58]. Overall, the role of HIF-2-mediated modifications in neural plasticity suggests that they may contribute to the brain’s adaptation to HA environments. However, more research is needed to fully understand the mechanisms underlying these effects and their potential therapeutic implications.

## 7. Potential Therapeutic Implications of Targeting HIF-2-Mediated Modifications in HA-Related Illnesses

Targeting HIF-2-mediated modifications may have therapeutic implications for HA-related illnesses. For example, HIF-2 inhibition may reduce the risk of developing AMS by preventing hypoxia-induced alterations in neurotransmitter systems and synaptic plasticity. Similarly, targeting HIF-2-mediated modifications may also help prevent HACE, a potentially fatal condition characterized by brain swelling and increased intracranial pressure. In chronic mountain sickness characterized by excessive erythrocytosis and pulmonary hypertension, targeting HIF-2-mediated modifications may also be beneficial. HIF-2 inhibition may help reduce erythrocytosis by decreasing erythropoietin production and preventing excessive hematopoiesis. Additionally, targeting HIF-2-mediated modifications may help reduce pulmonary hypertension by preventing hypoxia-induced changes in vascular tone and endothelial function. It is important to note that HIF-2 has various functions in different cell types and physiological conditions. Therefore, any therapeutic strategy targeting HIF-2 should be carefully designed to avoid potential side effects. Possible approaches could be:Targeting HIF-2-mediated pathways of neural plasticity may offer novel approaches for enhancing cognitive function and memory formation in HA environments, which can be impaired by hypoxia-induced changes in brain function.Gene therapy approaches targeting specific HIF-2-regulated genes or signaling pathways may provide a more targeted and long-lasting way to modulate the effects of hypoxia on the brain and improve HA-related illness outcomes.Developing personalized medicine approaches that consider an individual’s genetic and epigenetic profiles may help identify those at higher risk for HA-related illnesses and tailor interventions accordingly.Investigating the potential use of non-pharmacological interventions, such as cognitive or physical training, to enhance HIF-2-mediated modifications that promote neural plasticity and cognitive function in HA environments.

## 8. Other Factors than Hypoxia Might Affect the Brain at HA

While hypoxia is the primary driver of molecular changes in the brain at HA, other environmental factors such as hypobaria and hypothermia can also contribute to HA-related illnesses and affect brain function. Hypobaria can exacerbate hypoxia by reducing oxygen availability in the air, leading to more severe hypoxia and greater molecular changes in the brain. Additionally, hypobaria can affect the permeability of the blood-brain barrier and alter neurotransmitter signaling, which can contribute to HA-related illnesses such as AMS and HACE. Hypothermia can also affect brain function at HA. Exposure to cold temperatures can impair cognitive function and alter neurotransmitter systems, particularly dopamine and serotonin, which are involved in motivation and mood regulation. Additionally, hypothermia can exacerbate the effects of hypoxia by increasing oxygen consumption and decreasing oxygen delivery to the brain. While the molecular mechanisms underlying the effects of hypobaria and hypothermia on brain function at HA are less well-understood than those related to hypoxia, these factors can interact with hypoxia to exacerbate HA-related illnesses and affect brain function.

## 9. Conclusions

In summary, the brain undergoes several morphological and metabolic changes to adapt to reduced SpO_2_ at HA. Stabilization of HIFs may be the major mechanism in adapting to HA. HIF-2, especially, seems to be involved in hypoxia-driven activation of the glomus cells of the carotid body, angiogenesis, neurogenesis, survival, synaptogenesis, and brain metabolism, ultimately stimulating cognition in rodents and humans. Moreover, cerebral EPO is a crucial hormone in regulating the brain to maintain neuronal networks, working properly when challenged by hypoxia, and preventing cognitive deficits (Figure 2). The knowledge acquired from acute or prolonged hypoxia exposure experiments or HI models provides insights that may help us to predict how the brain adapts to HA. However, to fully understand the mechanisms and functions of HIFs, experiments are needed specifically designed to address HA-adaptive processes in the brain. Facilities, e.g., in La Paz, Bolivia (3600 m) (contact edith.schneidergasser@uzh.ch) or the Jungfraujoch Research Station (3500 m), Switzerland (contact markus.thiersch@uzh.ch) are available to investigate among other things the brain‘s adaptation to HA. Understanding the molecular mechanisms underlying the brain adaptation to HA may help to develop therapeutic approaches to mitigate the adverse effects of HA exposure. In particular, the EPO and HIF-2a pathways may be critical targets that open doors for researching innovative treatments for neuropsychiatric diseases.

## Figures and Tables

**Figure 1 ijms-24-10179-f001:**
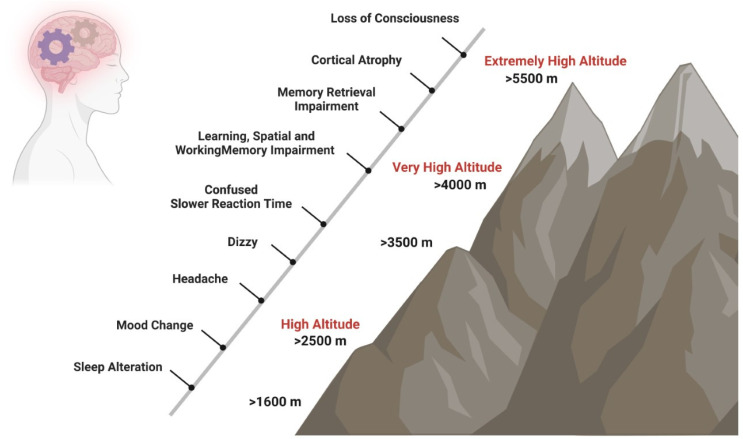
Impact of HA on cognition. Short- and long-term exposure to HA has effects on neurocognitive functions. Sleep patterns can be affected at elevations above 1600 m, changes in mood states like euphoria or depression are observed in some individuals from 2500 m on, and above 3000 m, subjects can experience headache, dizziness, and confusion. At very HA, slower reaction times, such as reduced attention (>3500 m), impaired learning, spatial and working memory (>4000 m), and impaired retrieval (>5500 m) may occur.

**Figure 2 ijms-24-10179-f002:**
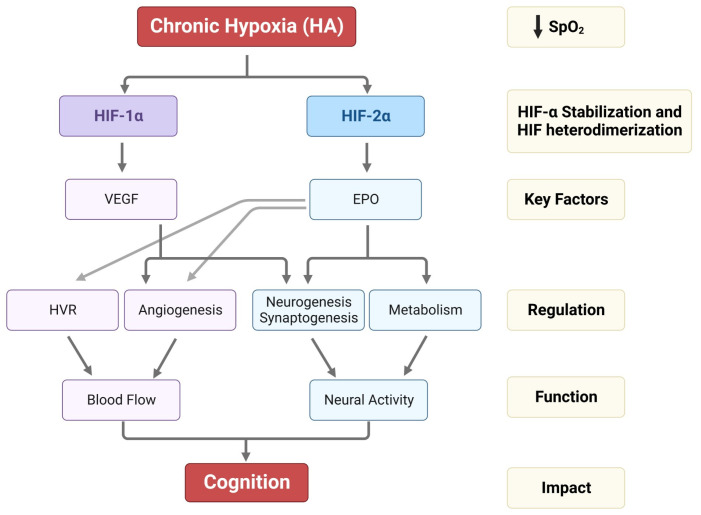
Impact of HA on HIF-1α, HIF-2α stabilization, VEGF and EPO expression, and their regulation in HVR, angiogenesis, neurogenesis, synaptogenesis, and metabolism. The alterations in HVR, angio-neurogenesis, and metabolism influence brain function and cognition.

## Data Availability

Data sharing not applicable.

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
