# Peer review of "The Brain at High Altitude: From Molecular Signaling to Cognitive Performance"

_ijms, 2023, doi:10.3390/ijms241210179_

Round 1
Reviewer 1 Report
The manuscript reads well and will add significant value to the field of neuroscience. I recommend acceptance of the manuscript following minor improvements.
Improve the grammar for a broad readership.
Provide primary references and appropriate abbreviations throughout the manuscript.
Elaborate on different mood states and their changes at high altitudes.
Author Response
Point 1: Improve the grammar for a broad readership.
Response 1: We sent the article to be proofread by an English speaker. If needed we will send the article to proofread again.
Point 2: Provide primary references and appropriate abbreviations throughout the manuscript.
Response 2: We try to provide as many primary references as possible. Some reviews are remaining. We will remove them is requested. We introduced all the abbreviations to the manuscript.
Point 3: Elaborate on different mood states and their changes at high altitudes
Response 3: Mood states at high altitudes were introduced in section 1 (see yellow marked paragraph page 1: 39-43 and page 2, 53-68).
Reviewer 2 Report
Such review may be very interesting and have clinical applications. Unfortunately, there is a large gap between the title and the paper. It is speaking about cognitive performances, when there is nothing about it. Knowing the brain physiology, makes easy to anticipate that the modifications triggered by high altitude, will affect the cognitive processes. But the paper does not present any evidence or any clinical data , that support this title. The authors must revise their paper with including some additional references (e.g. Pun et al., 2018; Bahrke and Shukitt-Hale 1993; Banderet and Burse 1991Shukitt-Hale and Liebermann, 1996; ...). All those authors have presented a description of cognitive and mood alterations. Another option is to go deeper into the discussion of the description of psychophysiological alterations, described at neuronal, membranary or synaptic level, and to discuss the consequences on cognitive processes.
Author Response
Point 1: Such review may be very interesting and have clinical applications. Unfortunately, there is a large gap between the title and the paper. It is speaking about cognitive performances, when there is nothing about it.
Response 1: We expanded the section on cognitive performances at HA in Chapter 1. (New paragraphs are labeled in yellow, page 1: 39-43, and pages 2 to 3: 53-102). Figure 1 has a new legend.
Point 2: Knowing the brain physiology, makes easy to anticipate that the modifications triggered by high altitude, will affect the cognitive processes. But the paper does not present any evidence or any clinical data , that support this title.
Response 2: This review paper aims to compare the effects of acute, and long-term high-altitude exposure on the cognitive function of lowlanders as well as alterations in cognition in highlanders. We aimed to put weight on the underlying physiological mechanisms, to provide a scientific basis for the assessment and protection of cognitive function.
Point 3: The authors must revise their paper with including some additional references (e.g. Pun et al., 2018; Bahrke and Shukitt-Hale 1993; Banderet and Burse 1991 Shukitt-Hale and Liebermann, 1996; ...). All those authors have presented a description of cognitive and mood alterations.
Response 3: Pun et al, 2018 (Reference 13) and Bahrke and Shukitt-Hale, 1993 (Reference 6) were already cited in the manuscript. Shukitt-Hale, Banderet, 1991 (Reference 11) and Shukitt-Hale and Liebermann, 1996 (Reference 10) were added.
Point 4: Another option is to go deeper into the discussion of the description of psychophysiological alterations, described at neuronal, membranes or synaptic level, and to discuss the consequences on cognitive processes.
Response 4: Chapter 5.4 discusses how HIF regulates neurotransmitters in neurons. Chapter 5.5 was added (Page 13:584-608) to discuss membrane alterations in neurons undegoing hypoxia.
Reviewer 3 Report
The review is devoted to the consideration of the mechanisms underlying changes in brain function during hypoxia. The review is of great interest. Excellent and logical presentation of the material reflects the current state of research in the field of brain function at high altitudes. The study is recommended for publication after minor corrections:
Minor poins:
Acute hypoxia significantly changes the intracellular redox balance, which underlies changes in a large number of proteins functioning. What is known about the antioxidant system of neurons and other cells of people at high altitude (highlanders and ordinary people after long-term adaptation)?
The study mentions the role of nitrosothiols in response to hypoxia. The redox modifications such as glutathionylation and nitrosylation are closely related to the regulation of protein function. Inhibition of NO synthase during hypoxia and the growth of oxidized glutathione level can lead to the induction of protein glutathionylation, which affects a number of proteins, including ion-transporters and HIF-1alpha.It would be great to mention about this modification under conditions of hypoxia.
Hypoxia tolerant Spalax specie are able to maintain redox status even under hypoxic conditions. What is known about role of HIF-1alpha and HIF-2alpha in the redox regulation of hypoxia tolerant species?
In the section about the mechanisms of animal adaptation, one can add about the role of membrane potential dissipation in the neuron functionality and cell damage during hypoxia and about channel arrest in hypoxia-resistant animals.
Author Response
Point 1: Acute hypoxia significantly changes the intracellular redox balance, which underlies changes in a large number of proteins functioning. What is known about the antioxidant system of neurons and other cells of people at high altitude (highlanders and ordinary people after long-term adaptation)?
Response 1: The exact mechanisms by which hypoxia affects ROS generation in neurons at high altitudes are not yet fully understood and may vary depending on the duration and severity of the hypoxic exposure. Research on animals and humans exposed to high altitudes has demonstrated changes in antioxidant enzyme activities and antioxidant molecule levels in the brain [244]. These adaptations are believed to be an essential part of the cellular response to counteract the increased ROS production and maintain neuronal function. Neurons are particularly vulnerable to oxidative stress due to their high metabolic activity, high content of oxidizable fatty acids, and relatively low levels of antioxidant enzymes compared to other cell types [245]. However, neurons have developed an antioxidant system to counterbalance ROS and maintain cellular redox balance. Some of the key components of the neuronal antioxidant system include several antioxidant enzymes, such as superoxide dismutase (SOD), catalase, and glutathione peroxidase; non-enzymatic antioxidants to scavenge ROS, including molecules such as vitamin C, vitamin E, glutathione, and uric acid; and various transcription factors, including nuclear factor erythroid 2-related factor 2 (Nrf2) that controls the expression of multiple antioxidant genes and helps maintain redox homeostasis [246]. (The paragraph is marked in yellow in the text. Page 11:491-506.
Point 2: The study mentions the role of nitrosothiols in response to hypoxia. Redox modifications such as glutathionylation and nitrosylation are closely related to the regulation of protein function. Inhibition of NO synthase during hypoxia and the growth of oxidized glutathione levels can lead to the induction of protein glutathionylation, which affects a number of proteins, including ion-transporters and HIF-1alpha.It would be great to mention about this modification under conditions of hypoxia.
Response 2
Glutathionylation is crucial in redox signaling and cellular adaptation to oxidative stress. It is an important mechanism in regulating the activity of various proteins, including ion transporters and hypoxia-inducible factor 1 (HIF-1) pathway proteins. Sodium/Potassium ATPase (Na+/K+-ATPase) is an essential membrane protein responsible for maintaining the electrochemical gradient across the plasma membrane of cells. Hypoxia-induces glutathionylation of specific cysteine residues in the α-subunit of Na+/K+-ATPase. This modification can alter the activity of the pump, affecting ion balance and cellular function [112, 113]. Calcium ATPase (SERCA) is an ATP-dependent calcium pump located in the endoplasmic reticulum (ER) membrane, involved in calcium homeostasis. Glutathionylation of SERCA occurs under hypoxia, leading to altered calcium handling and ER stress [114]. Glutathionylation of specific cysteine residues in HIF-1α can stabilize the protein, preventing its degradation and allowing it to accumulate and translocate to the nucleus for gene transcription [119]. Glutathionylation can also affect the DNA-binding activity of HIF-1α. It has been suggested that glutathionylation of specific cysteine residues in HIF-1α may enhance its binding to the hypoxia response elements (HREs) present in the promoter regions of target genes, thereby influencing their transcriptional activation [121]. Overall, glutathionylation of proteins, including ion transporters and HIF-1, is a crucial mechanism in the cellular response to hypoxia. It allows cells to modulate the activity of proteins involved in ion balance, calcium signaling, and gene expression, enabling adaptation and survival under conditions of oxygen limitation. Further research to fully understand the specific cysteine residues targeted for glutathionylation and the functional consequences of this modification on protein activity in the context of hypoxia is needed. (Paragraph in text highlighted yellow. Page 7:277-299).
Point 3. Hypoxia-tolerant Spalax specie are able to maintain redox status even under hypoxic conditions. What is known about role of HIF-1alpha and HIF-2alpha in the redox regulation of hypoxia tolerant species?
Response 3.
HIF-1α and HIF-2a are constitutively expressed at high levels in various tissues of hypoxia-tolerant species like Spalax, even under normoxic conditions. This elevated baseline expression is believed to preserve redox balance and energy metabolism even in low-oxygen environments [228]. (Paragraph yellow in text, P10:468-471)
Point 4. In the section about the mechanisms of animal adaptation, one can add about the role of membrane potential dissipation in neuron functionality and cell damage during hypoxia and about channel arrest in hypoxia-resistant animals.
Response 4. An entire section 5.5. "Membrane potential and ion exchange" has been added to the manuscript (Page 13: 592-616)
Round 2
Reviewer 2 Report
You have improved and clarified the parts of the paper, that were problematic. The paper is now easier to read and the relationship between title and contain is much improved. this is an interesting paper.
Author Response
We are grateful to the reviewer for the appreciation of the value of our work. We thank you very much for providing key suggestions and comments to better correlate the content with the title.